# Intrinsic Shapes of Empathy: Functional Brain Network Topology Encodes Intersubjective Experience and Awareness Traits

**DOI:** 10.3390/brainsci12040477

**Published:** 2022-04-05

**Authors:** Sjoerd J. H. Ebisch, Andrea Scalabrini, Georg Northoff, Clara Mucci, Maria Rita Sergi, Aristide Saggino, Antonio Aquino, Francesca R. Alparone, Mauro Gianni Perrucci, Vittorio Gallese, Simone Di Plinio

**Affiliations:** 1Department of Neuroscience, Imaging and Clinical Sciences (DNISC), G. d’Annunzio University of Chieti-Pescara, 66100 Chieti, Italy; antonio.aquino@unich.it (A.A.); francesca.alparone@unich.it (F.R.A.); maurogianni.perrucci@unich.it (M.G.P.); simone.diplinio@unich.it (S.D.P.); 2Institute of Advanced Biomedical Technologies (ITAB), G. d’Annunzio University of Chieti-Pescara, Via Luigi Polacchi 11, 66100 Chieti, Italy; 3Department of Psychological, Health and Territorial Sciences (DiSPuTer), G. d’Annunzio University of Chieti-Pescara, 66100 Chieti, Italy; andrea.scalabrini@unich.it; 4The Royal’s Institute of Mental Health Research, University of Ottawa, Ottawa, ON K1N 6N5, Canada; georg.northoff@theroyal.ca; 5Brain and Mind Research Institute, Centre for Neural Dynamics, Faculty of Medicine, University of Ottawa, Ottawa, ON K1N 6N5, Canada; 6Mental Health Centre, Zhejiang University School of Medicine, Hangzhou 310030, China; 7Centre for Cognition and Brain Disorders, Hangzhou Normal University, Hangzhou 310030, China; 8TMU Research Centre for Brain and Consciousness, Shuang Hospital, Taipei Medical University, Taipei 110, Taiwan; 9Graduate Institute of Humanities in Medicine, Taipei Medical University, Taipei 110, Taiwan; 10Department of Human and Social Sciences, University of Bergamo, 24129 Bergamo, Italy; clara.mucci@unich.it; 11Department of Medicine and Aging Sciences, G. d’Annunzio University of Chieti-Pescara, 66100 Chieti, Italy; mariaritasergi@libero.it (M.R.S.); aristide.saggino@unich.it (A.S.); 12Unit of Neuroscience, Department of Medicine and Surgery, University of Parma, 43121 Parma, Italy; vittorio.gallese@unipr.it

**Keywords:** empathy, intersubjectivity, functional magnetic resonance imaging fMRI, individual differences, brain networks

## Abstract

Trait empathy is an essential personality feature in the intricacy of typical social inclinations of individuals. Empathy is likely supported by multilevel neuronal network functioning, whereas local topological properties determine network integrity. In the present functional MRI study (N = 116), we aimed to trace empathic traits to the intrinsic brain network architecture. Empathy was conceived as composed of two dimensions within the concept of pre-reflective, intersubjective understanding. Vicarious experience consists of the tendency to resonate with the feelings of other individuals, whereas intuitive understanding refers to a natural awareness of others’ emotional states. Analyses of graph theoretical measures of centrality showed a relationship between the fronto-parietal network and psychometric measures of vicarious experience, whereas intuitive understanding was associated with sensorimotor and subcortical networks. Salience network regions could constitute hubs for information processing underlying both dimensions. The network properties related to empathy dimensions mainly concern inter-network information flow. Moreover, interaction effects implied several sex differences in the relationship between functional network organization and trait empathy. These results reveal that distinct intrinsic topological network features explain individual differences in separate dimensions of intersubjective understanding. The findings could help understand the impact of brain damage or stimulation through alterations of empathy-related network integrity.

## 1. Introduction

“*To perceive is to suffer*”—*Aristotle*

Trait empathy is a key notion in the study of individuals’ social inclinations originating in philosophical and phenomenological studies [1,2,3,4,5]. Although no unequivocal consensus has been reached in the literature about the exact definition of empathy, from a traditional phenomenological account, empathy can be summarized as the intersubjective understanding of others’ experiences [6,7]. Within this view, empathy is presumed to be characterized by at least two different dimensions: firstly, a perceptual component consists of experiencing the feelings of another individual through a vicarious process, which nevertheless appreciates self-other distinction; secondly, a cognitive aspect entails the natural awareness of other individuals’ experiences. These dimensions share some similarities with the distinction between sharing and understanding others’ emotions as commonly conceived in neuroscientific research [8,9,10,11,12], but also retain key conceptual differences [13]. In the present study, trait empathy is considered as a compound product of these constructs to describe individual differences within a common context of intersubjectivity.

The neural correlates of individual differences in trait empathy were studied by analyzing the relationships between its psychometric measurement and intrinsic brain properties. Such brain properties are believed to contribute to differences in the individual predisposition for empathic experiences. For instance, previous magnetic resonance imaging (MRI) studies focused on the association between local grey matter volumes and the subscales of the Interpersonal Reactivity Index (IRI [14]) or the Questionnaire of Cognitive and Affective Empathy (QCAE [15]). Functional MRI (fMRI) studies provided insights into the relationship of IRI subscales with intrinsic functional connectivity patterns during resting states in the absence of external task demands [16,17,18] or fractional amplitude of low-frequency fluctuations [19]. These studies linked a multiplicity of regions and networks that differed between affective and cognitive aspects of empathy. Notwithstanding the heterogeneity of the results, which may be attributed to the use of multiple empathy and brain measures and to an a priori selection of brain structures, these studies started to clarify an intrinsic neural basis of the various dimensions of empathy traits.

Some methodological issues need to be considered to advance the understanding of how intrinsic brain features contribute to individual differences in empathy. On the neural level, the brain can be considered a complex network with integration and segregation as functional principles that allow efficient information propagation to support cognitive functions [20,21]. To support different psychological functions, specific brain structures may play a crucial role in the information propagation through the brain network. This may be especially true for so-called hubs, i.e., structures that putatively facilitate the integration of neural subsystems [22]. Indeed, the consequences of brain damage go beyond local effects by altering global brain network functioning, which can lead to behavioral variation [23,24,25]. Although brain damage often compromises complex psychological functions such as empathy [26], the network architecture of trait empathy remains poorly understood. Its study is therefore important to further clarify the neural mechanisms as well as the possible impact of brain lesions [27], neurological disease [28,29], or brain stimulation [30,31,32] on empathy.

On a psychometric and theoretical level, the perceptual–cognitive dimensions of empathy considered in trait measurements are seldom conceived within a common theoretical framework. Although a common distinction concerns that between emotional [27,33] and cognitive empathy [19,34], these concepts are rooted in distinct theoretical backgrounds (e.g., “embodied simulation” versus “theory-theory” approach) [35,36]. Furthermore, it is disputed if the dimensions of the most frequently used empathy questionnaires (e.g., IRI [37]) can be integrated into perceptual–cognitive bipolarity [38]. Finally, psychometric questionnaires often assess emotional empathy as general responses to the emotional experience of someone else (e.g., “feeling for” instead of “feeling with” others’) and cognitive empathy as an effortful and explicit understanding of the emotions of another individual (based on cognitive inference rather than intuitive comprehension) [11,39,40]. Thus, this literature, as well as previous studies on the brain correlates of individual differences in empathy traits [16,17,18,19], mainly addresses a distinction between more general emotional responses and cognitive/reflective aspects of empathy, whereas the differentiation between perceptual and cognitive dimensions of empathy on a pre-reflective level (i.e., awareness without explicitly inferring others’ experiences) and their neural substrates have been relatively ignored [13].

In order to address these methodological, psychometric and theoretical issues, the present study aims at investigating the relationship between the network properties of whole-brain intrinsic functional connectivity profiles and psychometric measures of pre-reflective intersubjective empathic traits in a large sample. The brain network can be formally described as a graph, which comprises sets of vertices or nodes (neuronal elements such as brain regions) and edges (their interconnections). The pairwise couplings between brain regions are summarized in the network’s connectivity matrix, whose arrangement defines its graph topology [41,42]. By associating graph theoretical measures with individual scores in trait empathy dimensions, we can identify the contribution of brain regions to information propagation within empathy-related brain network structures.

Centrality measures indicate the importance of a node or a set of nodes for communication performance within a network [43,44]. Such indices are relevant to obtain insights, for example, on the intrinsic neural basis of empathy as well as on the impact of lesions on empathy, privileging the network perspective rather than localized functions. Since centrality can be defined in various ways, three commonly used centrality measures were used here to quantify the topological importance of brain regions. Firstly, degree centrality is determined by the number of connections of each node. Secondly, betweenness centrality is based on the number of shortest paths between pairs of nodes that pass through a determined node. Thirdly, closeness centrality depends on the average length of shortest paths between a node and all the other network nodes. Using many complementary measures of centrality allows the thorough description of nodal (and modular) contribution to the overall information flow within a network [45].

For the measurement of trait empathy, we used the Empathic Experience Scale (EES), which assesses empathy as a multidimensional construct with satisfying psychometric properties within a common framework of intersubjectivity based on bodily resonance (i.e., “vicarious experience”) and pre-reflective awareness (i.e., “intuitive understanding”) of others’ feelings [13]. Given that the gender differences in empathy and related brain networks have been consistently reported throughout the lifespan [46,47], we considered sex as a factor to investigate whether trait empathy could rely on different neural mechanisms in males and females.

Although emotional and cognitive empathy are considered distinct processes largely associated with differential brain circuits, interpersonal sharing and pre-reflective understanding are often considered intertwined processes. It is still unclear whether and how different brain systems have distinguishable roles in vicarious experience and intuitive understanding as parts of pre-reflective forms of empathy [48,49,50]. In accordance with evidence that vicarious experience and intuitive understanding reflect psychometrically independent dimensions [13], we hypothesized that individual differences in these distinct dimensions are associated with connectivity profiles of distinct brain regions within empathy-related networks.

Brain hub regions that have been linked with affective or cognitive aspects of empathy include the precuneus, anterior cingulate cortex, anterior insula, ventromedial and superior frontal cortex, superior and inferior parietal areas [10,22,26,51,52]. These regions greatly overlap with the salience network [53], the default mode network [54], the fronto-parietal network [55] and brain systems involved in self-related processing as well as in personal emotional experiences [56,57]. Shared circuits for the processing of self- and other-related information additionally include (pre)motor and somatosensory cortices as part of the mirror neuron system (MNS), which overlaps with fronto-parietal and somatomotor networks [58,59,60,61].

Based on the above literature, the fronto-parietal network, including the MNS, implied in the empathic sharing of actions, emotions, and sensations [58,61] could be expected to explain individual differences in the vicarious experience trait. By contrast, the intuitive understanding trait partially resembles the functions ascribed to default mode, prefrontal and temporo-parietal junction regions [10,11], although it particularly differs from cognitive empathy through its emphasis on the absence of effortful inferential processes. Finally, the salience network could show overlapping associations with both vicarious experience and intuitive understanding, constituting a common system for multiple empathic processes [51,52]. However, we expect that the network properties of brain regions related to vicarious experience and intuitive understanding may reasonably differ from those typically related to similar constructs in the literature as described above. For example, vicarious experience and intuitive understanding show moderate to low correlates with other psychometric measures frequently used in neuroimaging to study emotional and cognitive empathy, suggesting that the EES scores are complementary to other questionnaires due to different construct definitions [13].

Moreover, considering the focus of the present study on the brain network architecture, we hypothesized a partial overlap with brain structures implied by neuropathology since brain damage likely disrupts network functioning in addition to local brain function. For instance, in accordance with the studies in patients with prefrontal lesions [62], impaired emotional empathy is also an early, central symptom of frontotemporal dementia [28,29,63], a condition commonly associated with early atrophy in orbitofrontal and medial/lateral prefrontal cortices. Moreover, subcortical structures were identified as main hub regions [64,65] with evolutionary importance in empathy [57,66], and their lesions likely interrupt networks relevant for the empathic recognition of emotions [67,68,69]. Hence, connectivity profiles of medial/lateral prefrontal cortices and subcortical structures could be particularly relevant for intuitive understanding traits.

## 2. Materials and Methods

### 2.1. Participants

One hundred and sixteen healthy Italian adults (62 females, of which 60 were right-handed, and 54 males, of which 52 were right-handed; aged 23 ± 3) without a history of psychiatric or neurological disease and contraindications for MRI scanning participated in the experiment. The experiment was approved by the local ethics committee. All participants had a normal or corrected-to-normal vision and provided written informed consent before taking part in the study in accordance with the Declaration of Helsinki (2013). Participants partially overlapped with those included in Di Plinio et al. [70] (N = 38). All the participants performed two resting-state runs for assessing the functional architectures of the brain and filled the Empathic Experience Scale [13] for the assessment of trait empathy.

### 2.2. Empathic Experience Scale

The EES is a self-report questionnaire composed of 30 items to evaluate pre-reflective intersubjective understanding within a bidimensional empathy model. It provides individual scores on two subscales: vicarious experience and intuitive understanding. According to the original publication of the questionnaire [13], the EES provides a reliable and valid measure of empathy characterized by a two-factor structure with reported internal consistencies of 0.93 and 0.95 for vicarious experience and intuitive understanding, respectively. Vicarious experience concerns an individual’s inclination to experience emotions similar to the internal state of another individual. Intuitive understanding regards the predisposition to naturally, not effortfully, recognize the emotional state of another individual. All participants completed the EES in a quiet room on a different day than the fMRI scanning after receiving the following standardized instructions: “Please read very carefully the following statements and rate how strongly they describe how you normally feel. Use the following scale. Do not linger too much on the single statements and answer as sincerely as you can”. The scoring is based on the responses provided by the participant on the items on a Likert scale from 1—“Not at all true” to 5—“Completely true”. The vicarious experience score is obtained by summing up all the responses to the 15 odd items, whereas the intuitive understanding score is obtained by summing up all the responses to the 15 even items. No items need to be inverted.

### 2.3. MRI Data Acquisition and Preprocessing

Imaging data were acquired using a 3 Tesla Philips Achieva MR scanner at the Institute of Advanced Biomedical Technologies (ITAB) in Chieti, Italy. Head motion was minimized using foam padding and surgical tape. Participants were instructed to avoid movements and avoid falling asleep. The total sample (N_TOT_ = 116) was composed of four subgroups that participated in four different experiments. Thus, some parameters changed across scanning procedures. We took care of this aspect in the analysis by adding a random grouping for the “scanning group” in the regression models.

For the first group (N_I_ = 38 participants [70]), each participant performed two consecutive resting-state fMRI runs, each consisting of 376 volumes, during which they watched a white fixation cross on a black screen. Whole-brain functional images were acquired with a gradient echo-planar sequence using the following parameters: repetition time (TR) = 1.2 s, echo time (TE) = 30 ms, field of view (FoV) = 240 mm, flip angle = 65°, in-plane voxel size = 2.5 mm^2^, slice thickness = 2.5 mm. A high-resolution T1-weighted whole-brain image was also acquired after functional sessions using the following parameters: TR = 8 ms, TE = 3.7, FoV = 256 mm, flip angle = 8°, in-plane voxel size = 1 mm^2^, slice thickness = 1 mm.

For the second (N_II_ = 28) and third (N_III_ = 22) groups, each participant performed two task-free fMRI runs, each consisting of 234 volumes. Whole-brain functional images were acquired with a gradient echo-planar sequence using the following parameters: TR = 1.8 s, TE = 30 ms, FoV = 240 mm, flip angle = 85°, in-plane voxel size = 3 mm^2^, slice thickness = 3.5 mm. A high-resolution T1-weighted whole-brain image was also acquired after functional sessions using the following parameters: TR = 8 ms, TE = 3.7 ms, FoV = 240 mm, flip angle = 8°, in-plane voxel size = 1 mm^2^ and slice thickness = 1 mm.

For the fourth group (N_IV_ = 28), each participant performed a task-free fMRI runs, consisting of 605 volumes. Whole-brain functional images were acquired with a gradient echo-planar sequence using the following parameters: TR = 1.0 s, TE = 35 ms, FoV = 230 mm, flip angle = 90°, in-plane voxel size = 2.875 mm^2^, slice thickness = 3.5 mm. A high-resolution T1-weighted whole-brain image was also acquired after functional sessions using the following parameters: TR = 8 ms, TE = 3.7 ms, FoV = 230 mm, flip angle = 8°, in-plane voxel size = 1 mm^2^ and slice thickness = 1 mm.

### 2.4. MRI Data Preprocessing

The preprocessing of fMRI and MRI images was implemented in AFNI software [71] (Analysis of Functional Neuroimages, https://afni.nimh.nih.gov, accessed on 4 July 2020). The quality and temporal organization of steps followed the updated standards suggested by AFNI developers and were common across all participants.

For each subject, the two resting-state runs were concatenated. The resulting functional images were despiked, deobliqued and time-shifted to obtain slices with the same acquisition timing. The high-resolution anatomic image (T1) was skull-stripped, aligned to the Montreal Neurological Institute (MNI) template, motion-corrected using a six-parameter rigid body realignment and then warped to the standard space through a non-linear high-efficiency procedure. Thus, functional images were aligned to the standard space and to the T1 image using automatically optimized non-linear warping algorithms. During preprocessing, motion parameters for each participant were stored to be used as a regressor of non-interest in further steps. The resulting functional images were blurred using a Gaussian smoothing kernel with 5 mm at its full-windows half maximum. Finally, fMRI time series were scaled to obtain an average value of 100, a procedure essential to translate the BOLD signal to a “percent signal change”, which is a more meaningful measure [72]. The quality of the preprocessing was assessed both using AFNI built-in quality check programs and utilities and through the direct examination of the preprocessed images.

Timeseries were additionally censored, meaning that volumes with 10% or more outliers across voxels were excluded. Furthermore, volumes with the Euclidean norm of the motion derivative exceeding 0.2 mm were also excluded [73]. Censoring and band-pass filtering in the frequency interval [0.01, 0.10 Hz] were applied in a single regression step as suggested by methodological studies [74]. In this step, motion parameters were included as noise regressors together with white matter and cerebrospinal fluid signals. The mean framewise displacement was also added as an additional covariate of no interest [73,75]. The global signal was not regressed because it is a controversial approach [76] and introduces non-trustable statistical associations across voxels [77].

### 2.5. fMRI Data Analysis

The two resting-state runs of each subject were concatenated to obtain whole-brain functional connectomes using the z-Fisher transform of the Pearson correlation among average time series extracted from the voxels within each node. Correlation matrices were obtained from a set of 418 nodes using cortical (346 nodes) and subcortical (40 nodes) atlases from Joliot et al. [78] plus the cerebellar (32 nodes) atlas from Diedrichsen et al. [79]. Graph analyses were performed within Matlab (The MathWorks, Inc., Natick, MA, USA, Version 2020a) using the Brain Connectivity Toolbox [80]. The software BrainNet Viewer [81] (https://www.nitrc.org/projects/bnv/, accessed on 1 June 2020) was used to visualize modular structures. Binary undirected graphs were built from connectomes (connectivity matrices) after a thresholding procedure in which the 10% among the strongest connections were maintained to remove spurious connectional values [82,83] and to obtain sparsely connected matrices at the single-subject level with the aim of keeping the total number of connections fixed across all individuals. Such an approach controls for the influence of network density on the computation and comparison of graph metrics across individuals [84,85].

Modularity maximization [86] through the application of the robust Louvain algorithm [87,88] combined with an iterated fine-tuning algorithm was used to detect optimal modular architectures at a single-subject level [89] and to handle the stochastic nature of the Louvain algorithm [90]. The process works as follows: first, each node is assigned into a separate module; second, the robust Louvain algorithm is applied to detect the optimal community; third, the modularity Q is estimated; then, the procedure is repeated using the optimal community found in the previous cycle as the starting community in the current cycle. Such an algorithm is until the modularity cannot be incremented anymore. Once repeated the entire iterative fine-tuning process for each participant, an agreement matrix was calculated as the matrix whose elements how many times the two nodes were included in the same community (module) across participants. The agreement matrix allowed estimating the cross-subject (group-level) modular architecture; a community detection algorithm developed for the analysis of complex networks [91] was implemented to that aim, with 1000 repetitions to avoid sub-optimal results.

The structural resolution (γ) is crucial in module detection since it weights the null model when calculating modularity measures. Here, we varied γ in the range 0.3–5.0 with steps of 0.1 to implement an unbiased procedure [92]. The Newman–Grivan procedure was used to detect significant modules in the group-level architectures [93]. Values of modularity (Q) were extracted for each individual at each level of structural resolution investigated and reported [94]. By using a linear Pearson correlation, we tested if modularity was associated with vicarious experience and intuitive understanding, possibly reflecting a “global” encoding of empathic traits in the brain connectome.

In line with the empathy framework and with recent advances in network neuroscience, we wanted to study associations between within- and between-network exchange of information and empathic traits. Thus, we focused on metrics of centrality and participation. The centrality measures are diversified sets of measures useful to estimate the importance of nodes/modules in the information flow within the whole network [42]. We focused on degree centrality, closeness centrality and betweenness centrality. Degree centrality indicates the number of links connected to the node. Closeness centrality, instead, indicates how much a given node is close to every other node in the network. Lastly, betweenness centrality is the fraction of all shortest paths in the network that contain a given node and thus recapitulates how frequently nodes participate in the communication across other nodes. The participation coefficient measures the extent of intermodular connections of a node, thus representing the strength of cross-modular connections, whereas the within-module degree measures the extent of intra-modular connections of a node [95].

### 2.6. Analysis of Neural Networks-Empathic Traits Associations

Graph measures described above (degree centrality, closeness centrality, betweenness centrality, participation coefficient and within-module degree) were extracted in each node. The association of resting-state graph measures with empathic scores of vicarious experience and intuitive understanding was investigated using a two-step, robust linear mixed-effects modeling. At each value of structural resolution γ, a regression model was fitted for each behavioral score of interest using the module’s nodes as random groupings. Random intercepts were added at the subject level. Random slopes were included to detect both the modular-level estimate of the effects (fixed effect of the behavioral score) as well as the individual-node deviations from the estimate (random slopes) in the brain–behavioral associations while accounting for inter-individual variability at the subject level. Secondly, to check model assumptions, we repeated the analyses after the exclusion of nodes in which residuals were not homogeneously distributed as indicated by the Anderson–Darling normality test. The results were corrected for multiple comparisons using the false discovery rate (FDR).

To show significant results appropriately and meaningfully, we extracted nodal best linear unbiased predictors (BLUPs [96]) and produced individual conditional expectation plots (ICE plots [97]) to represent significant effects at both the modular and nodal level. We investigated the significance of brain–behavior associations at every structural resolution (γ) value to eliminate every possible source of selection bias. Coherently with what is indicated in the literature, age and sex (males, females) were included as covariates, and a random slope was added to control for sex-specific associations between nodal topology and empathy. Since it could be assumed that the acquisition procedure may differentially impact brain–behavior interactions, an additional random effect grouped for the scanning procedure was carried out. The usage of this random effect allows excluding the possibility that results may be due to inhomogeneity in the samples or in the scanning procedures. Whenever significant results (after FDR correction) were detected, the link between the brain and behavioral measures was tested in a multivariate model including all the structural resolutions in a single analysis step, with levels of γ being used as repeated measures. This model allowed a more detailed and interpretable inspection of the effects at the modular and the nodal levels. In the results section, statistics for these comprehensive models are reported in the text; statistics for the single-γ models are reported in the figures.

## 3. Results

### 3.1. Behavioral Results: Empathic Experience Scale

Vicarious experience (VE) and intuitive understanding (IU) were confirmed as independent and partially overlapping constructs (Figure 1). Both the subscales of the EES questionnaire showed high internal consistency: α_VE_ = 0.92; α_IU_ = 0.95. The two scales were low-to-moderately correlated (*r* = 0.315, *p* = 0.001, 95% CI [0.140, 0.470]). Paired *t*-tests showed that the scores of intuitive understanding were equal between males and females (*t*_114_ = 0.01; *p* = 0.99; confidence intervals for mean difference [−3.81, 3.85]). Instead, a significant difference was found when contrasting males vs. females with respect to their vicarious experience scores, with females showing higher scores (*t*_114_ = 4.85; *p* < 0.001; Confidence Intervals for mean difference [5.89, 14.03]). This suggests that females could have a stronger tendency than males to experience vicarious sensorimotor or emotional states when witnessing the emotional state of someone else, i.e., to participate stronger in other individuals’ emotional states by experiencing similar emotions. These results confirm findings from Innamorati and colleagues [13]. The two scales were transformed using Box–Cox transformation for the improvement of normality before being analyzed in mixed-effects models for brain–behavior correlations.

### 3.2. Global Modularity

Average and single-subject values of modularity (Q) across values of structural resolution (γ) are reported in Figure 2. As expected, values of optimized modularity decreased with increasing values of structural resolution (smaller modules). These values are consistent with the literature [70]. Modularity was not associated with vicarious experience (average cross-γ Person correlation = 0.014 ± 0.049, *p* values < 0.30) nor it was with intuitive understanding (average cross-γ Person correlation = −0.066 ± 0.022, *p* values < 0.30).

### 3.3. Brain Correlates of Vicarious Experience

As mentioned above (see Materials and Methods), we report here the results related to post hoc cross-structural resolution modules. The results for single levels of structural resolutions corrected for multiple comparisons are shown in Figure 3. Firstly, a positive association was detected between degree centrality and vicarious experiences in a module that encompassed the prefrontal cortex, intraparietal sulcus, inferior frontal cortex, precentral gyrus, anterior insula, orbitofrontal cortex and posterior temporal regions (Figure 3A; β = 3.2 ± 1.1; CI [1.1, 5.3]; *t* = 3.03; *p* = 0.0024). These regions largely overlap with the fronto-parietal network [55], which comprises, among others, also the MNS [61]. The fronto-parietal network (also known as the central executive network) shows strong coactivation during a wide range of cognitively demanding tasks and serves to rapidly instantiate new task-states by flexibly interacting with other networks such as the default mode network for introspective and self-related processing and the dorsal attention network for perceptual attention [98,99]. The examination of random slopes showed that such association was particularly strong in the dorsolateral prefrontal cortex and was absent or reverted in the dorsal anterior cingulate gyrus and supplementary motor cortex (BLUPs range [−3.8, 6.4]). No interaction was observed with sex (β_SEX_ = −2.1 ± 1.4; CI [−4.9, 0.7]; *t* = −1.47 *p* = 0.14).

Secondly, vicarious experiences were significantly associated with betweenness centrality (Figure 3B; β = 33 ± 16; CI [2, 65]; *t* = 2.1; *p* = 0.04). Similar to degree centrality, this relationship involved a module encompassing the cingulum, prefrontal, parietal, anterior insula, orbitofrontal and posterior temporal regions, similar to the fronto-parietal network described above. A significant interaction was observed with sex, showing that the relationship between betweenness centrality and vicarious experience in the fronto-parietal module was strong in females and was absent or slightly negative in males (β_SEX_ = −49 ± 23, CI [−93, −5], so that the partial effects were β_MALE_ = −19 and β_FEMALE_ = −82; *t* = −2.2; *p* = 0.03). The study of random slopes for the level positively involved in the prediction, that is, for females, showed that the significance was particularly high in cingulate and in dorsolateral prefrontal regions, especially in the right hemisphere.

Finally, a significant association was detected with participation coefficients in the fronto-parietal module encompassing prefrontal and orbitofrontal cortex, anterior insular cortex, posterior temporal nodes and mid-parietal nodes (Figure 3c; β = 0.016 ± 0.005; CI [0.006, 0.026]; *t* = 3.2; *p* = 0.0015). The examination of random slopes showed that different brain nodes had a different weight in such interaction: the effect was particularly strong in the dorsal anterior cingulate cortex and orbitofrontal regions (BLUPs range [0.0008, 0.0260]). No interaction was observed with sex, showing that the relationship between participation coefficients and vicarious experience was equal across males and females (β_SEX_ = −0.011 ± 0.007; CI [−0.025, 0.003]; *t* = 1.6; *p* = 0.11).

Regions of interest and their coordinates in the MNI space with respect to the brain correlates of vicarious experiences are reported in the Appendix A.

No significant results were observed with respect to measures of within-module degree nor with closeness centrality.

### 3.4. Brain Correlates of Intuitive Understanding

A significant positive association was found between degree centrality and intuitive understanding in a subcortical module encompassing thalamus, dorsal striatum (i.e., caudate and putamen) and globus pallidus (Figure 4A; β = 1.3 ± 0.5; CI [0.3, 2.2]; *t* = 2.6; *p* = 0.0086). A significant interaction was observed with sex, showing that the association was true for females and not for males (β_SEX_ = −1.6 ± 0.6, CI [−2.8, −0.3], so that the partial effects were β_MALE_ = −0.3 and β_FEMALE_ = 2.9; *t* = −2,5; *p* = 0.012). The effect on females was particularly strong in caudate and putamen regions.

Secondly, for betweenness centrality, a significant positive association with intuitive understanding was found in a somatomotor module encompassing the bilateral supplementary motor area, dorsal anterior and posterior cingulate cortex, precentral gyrus, central sulcus, anterior and posterior insula, and right striatum (Figure 4B; β = 25 ± 10; CI [5, 45]; *t* = 2.5; *p* = 0.017). Compared to the fronto-parietal network associated with vicarious experience, this network included more centrally located, primary sensorimotor areas. The interaction with sex was not significant (β_SEX_ = −20 ± 14; CI [−48, 8]; *t* = −1.4; *p* = 0.17). The significant association with betweenness centrality was stronger in cingulate regions of the module, while it was weaker in posterior somatomotor and mid-insular nodes.

Finally, a significant, positive association was detected between participation coefficients and intuitive understanding in a small module encompassing regions of the so-called salience network (Figure 4C; β = 0.014 ± 0.005; CI [0.004, 0.024]; *t* = 2.7; *p* = 0.0077). A significant interaction was observed with sex, showing that the relationship between participation coefficients and intuitive understanding in this salience module was present in females but absent in males (β_SEX_ = −0.015 ± 0.007, CI [−0.030, −0.001], so the partial effects were β_MALE_ = −0.001 and β_FEMALE_ = 0.031; *t* = −2.1; *p* = 0.035). As noticeable in the figure, the effect was particularly strong in the anterior insula.

Regions of interest and their coordinates in the MNI space with respect to the brain correlates of vicarious experiences are reported in the Appendix A.

## 4. Discussion

The present study aimed to investigate the relationship between the intrinsic functional architecture of the brain and trait empathy. On a behavioral level, we focused on the pre-reflective aspects of empathy within a common theoretical framework of intersubjectivity. Individual differences in the predisposition for empathy were assessed as composed of: (1) vicarious experience, i.e., experiencing the feelings of other individuals; (2) intuitive understanding, i.e., natural awareness of other individuals’ experiences. We performed whole-brain graph analyses to provide a new window into the intricated information propagation features of brain networks that contribute to individual differences in empathic inclinations.

Overall, positive associations between vicarious experience scores and the fronto-parietal network were detected for degree and betweenness centrality as well as participation coefficients. On the other hand, regarding intuitive understanding scores, a positive association was found with degree centrality in a subcortical network and with betweenness centrality in a somatomotor network. Closeness centrality was associated neither with the vicarious experience trait nor with the intuitive understanding trait. Intriguingly, the betweenness centrality of salience network regions, e.g., anterior insula and cingulate cortex, were associated with both vicarious experience and intuitive understanding.

### 4.1. The Fronto-Parietal Encoding of Vicarious Experience

“*I do not ask the wounded person how he feels, I myself become the wounded person.*”—*Walt Whitman, Song of Myself*

Vicarious experience trait was related to three graph metrics of the fronto-parietal network: degree centrality, betweenness centrality and participation coefficients. The main brain regions that contributed to this effect include the bilateral prefrontal cortex, premotor cortex, dorsal anterior cingulate cortex, inferior parietal lobule, left orbitofrontal cortex, supplementary motor area and anterior insula. The functions of the fronto-parietal network are usually attributed to working memory, sustained attention, task-relevant information maintenance [100] and flexible task control [101]. Notably, it also overlaps with the MNS (e.g., inferior frontal cortex, premotor cortex, inferior parietal cortex [56,60,61]. Brain regions of the MNS putatively allow establishing a vital link between self and others during social interaction through a sensorimotor resonance with others’ experiences and basic motor intentions [7,35,102]. In other words, the MNS is hypothesized to support a pre-reflective understanding through the vicarious experience of others’ bodily experiences, including emotions [35,48,49,58]. Such a multiplicity of cognitive functions of the fronto-parietal network could be backed by its characteristic flexible interactions with other networks [16,55,100]. The present findings suggest that the typical individual inclination to resonate with others’ emotions is shaped by the intrinsic topological features of MNS regions together with prefrontal and orbitofrontal regions. It should be added that recent findings in macaques revealed the presence of mirror neurons in the prefrontal cortex [103].

The involvement of the prefrontal cortices in vicarious experience adds to evidence for their involvement in inferential social cognitive processes [27]. Findings in patients with acquired brain lesions and frontotemporal dementia provide converging evidence for the importance of the fronto-parietal network for emotional empathy traits, including affective perspective-taking [26] and emotion recognition [104], especially through the integrity of regions in medial, lateral and orbitofrontal cortices [28,29,62,63]. Since these regions represent crucial hubs within the empathy network, brain damage likely impairs vicarious experiences by disrupting socioemotional information processing [23,24,25]. The finding that participation coefficients, but not a within-module degree, predicted vicarious experience trait means that cross-network information exchange, rather than within-network interactions, is particularly relevant for vicarious experiences.

The fronto-parietal network also contributes to emotion regulation [105] and self-awareness [56]. While vicarious experience implies the sharing of others’ emotions, self-awareness and emotion regulation are necessary processes for empathy to recognize who is the primary source of emotions and to avoid becoming overwhelmed by others’ emotions. The fronto-parietal network could facilitate the complex information exchange across brain networks relevant to linking internal and affective processing, generally associated with the default mode network [57,106,107], with external processing in sensorimotor systems for bodily action and perception [55]. It could be hypothesized that the convergence of self-related and other related information in the fronto-parietal network may regulate the interplay between self-other sharing and self-other distinction that typifies empathy [108]. Accordingly, Innamorati and colleagues [13] showed that items addressing self-other sharing and distinction converged psychometrically in the vicarious experience factor.

Further studies are needed to clarify whether and how self-other resonance and distinction rely on overlapping neural circuits in the fronto-parietal network. However, present evidence suggests that the fronto-parietal network already differentiates between self- and other-related information, both in terms of stronger activation for the self and only partial overlap between self and other [49].

### 4.2. Multiple Brain Systems Involved in Intuitive Understanding

“*It’s the hardest thing in the world to go on being aware of someone else’s pain.*”—*Pat Barker, Life Class*

The connectional correlates of intuitive understanding were more variable than those of vicarious experience. The degree of centrality of a subcortical module enclosing the caudate, putamen and pallidus nuclei was positively related to intuitive understanding scores. The dorsal striatum was related to ancient evolutionary functions supporting social behavior [66,109], including decision making in action selection based on the expected reward [110,111], which contributes to social behavior [112,113]. Although the specific role of the striatum in empathy remains elusive, subcortical lesions were reported to impair emotional empathy [67,68,69]. Since subcortical structures contain major hub regions in brain networks [64,65], the present findings suggest that subcortical lesions could disrupt empathy-related networks and consequentially worsen the intuitive understanding of others’ emotions.

Differently, the betweenness centrality measures were correlated with intuitive understanding scores in an extended sensorimotor network, including the bilateral supplementary motor area, dorsal anterior and posterior cingulate cortex, precentral gyrus, central sulcus, anterior and posterior insula, and right striatum. Primary sensorimotor circuits were regularly associated with affective components of empathy [51,114,115,116,117,118], putatively allowing a link between one’s own and others’ experiences of actions, sensations and emotions [7,35,58]. Different from the fronto-parietal network linked with vicarious experience, which included mostly higher-order sensorimotor regions, the sensorimotor regions associated with intuitive understanding encompassed many low-level motor regions. Resting-state intrinsic functional connectivity profiles of low-level sensorimotor circuits were previously associated with individual differences in empathic concern [16]. Moreover, a substantial overlap can be noted with an empathy network consisting of the dorsal anterior cingulate cortex, the supplementary motor area, and the bilateral insula, as yielded by some meta-analyses [51,119].

The anterior insula was proposed to have a key role in the understanding of others’ bodily feelings [40] through first-person interoceptive processing [120,121]. Moreover, the insula was found to take an important role in intersubjective interaction [122,123]. The anterior insula is also part of the salience network and, together with the dorsal anterior cingulate cortex, allows the coupling of interoceptive processing with the motor system to guide behavior [120,124]. The association of participation coefficients with intuitive understanding agrees with a role of the bilateral anterior insula, possibly together with the supplementary motor cortex [125], in mediating between-network interactions [126] to support social cognition [124].

### 4.3. Brain Networks and Empathy Concepts

Our findings show some correspondences with previous studies on empathy-related brain networks that consider the empathy concept within a bidimensional framework. A vast literature distinguished between perceptual/affective and cognitive aspects of empathy, with distinct but complementary functions in social cognition. Neuroimaging research often linked affective empathy with the somatomotor system and the salience network, which has relevant connections with limbic and subcortical structures, whereas cognitive empathy is supposed to be supported by the default mode network, the prefrontal cortices, including the orbitofrontal cortex, the temporal lobe and the temporoparietal junction [10,11,35,40,56].

However, there are essential differences between the literature and the present study. One of our aims was a rigorous focus on psychometric measures of (1) vicarious experiences, avoiding confusion with emotional responses that are dissimilar to others’ emotions, such as concern or sympathy, as well as (2) intuitive understanding, which is distinct from top-down controlled understanding through cognitive inference. Although neuroscientific studies commonly investigated reactions to others’ affective experiences, the neural processes underlying concern and vicarious experiences are difficult to disentangle in typical experimental paradigms exposing participants to others’ emotional experiences. Furthermore, differently from other instruments (e.g., IRI), the EES allows exploring empathy as a natural awareness of others’ feelings without explicit cognitive inference and without biasing the behavioral measures with affective responses to others as feelings of concern or distress. The pre-reflective perceptual and cognitive components of empathy are frequently blended in conceptions of emotional/affective empathy that imply an implicit understanding through emotional resonance [48,127,128]. The present findings, therefore, expand previous knowledge by showing a different neurofunctional basis of the two pre-reflective sub-dimensions of empathy, i.e., perceptual (vicarious experience) and cognitive (intuitive understanding) aspects that parallel their psychometric properties [13].

On the one hand, the association of vicarious experience with two centrality measures in the fronto-parietal system indicates a multifaceted involvement of these fronto-parietal nodes in processes of self-other resonance, encompassing both radial (degree) and medial (betweenness) information flow [44,45]. In fact, these nodes have a role both in the propagation of information across subsystems (probably through their high inter-network connectivity [55]) and in the high-order modulation of socioemotional and affective facets of precepted stimuli [24,25,26]. On the other hand, intuitive understanding relies on different brain systems. Thus, the natural awareness of others’ feelings is influenced simultaneously by the *connectional strength* (degree) of subcortical regions and by the *connectional design* of sensorimotor regions, supporting the importance of both systems in empathic processes [16,67,68,114].

Importantly, the involvement of salience network regions’ betweenness in both empathic dimensions (vicarious experience and intuitive understanding) indicates a convergent engagement of these brain regions in self-other matching processes, including experience as well as an understanding of one’s own and others’ emotions and feelings [129,130,131,132,133]. The anterior insula was associated with different levels of self-processing from interoceptive to exteroceptive and mental levels [134], showing higher resting-state centrality indices for all three levels of self [135]. Our findings imply that the anterior insula could be a key hub for the propagation of general viscero-motor information necessary for sharing as well as understanding others’ emotional states [51,136]. Removing anterior insular or dorsal cingulate nodes from the network would impair the merging of self-related information with the representation of others’ feelings which is necessary for bodily resonance and pre-reflective understanding of others. These findings help to explain behavioral disturbances highlighted in lesion studies of the anterior insula and cingulate cortex, such as alexithymia [137], aberrant interpersonal behavior [138] and decreased social interactions [139]. Overall, we further extended the functional role of the insula, which could act as a key hub not only for self-processing (subjectivity) but also for empathic processing (intersubjectivity).

### 4.4. Sex Effects

We found that the neural representation of empathy traits is different across females and males. In many cases, the observed associations tended to be stronger for females compared to males. For example, a stronger relationship between vicarious experience and betweenness centrality of the fronto-parietal network was found in females. Future studies should clarify whether such a relationship contributes to the observed higher vicarious experience scores in females compared to males, as also reported by previous studies [13,140,141,142]. It could be hypothesized that the tendency for a higher vicarious experience score among women could be explained in the light of their role as primary caregivers for children. This inclination may have favored selective pressures for greater vicarious empathy in women with the benefit of reading the infant’s mental states and needs rapidly, thus promoting the infant’s survival [143,144].

In contrast to vicarious experience, intuitive understanding scores tend to be similar between males and females [13]. Nevertheless, interactions between sex and participation coefficients showed that a role of the insula and supplementary motor cortex in mediating between-network interactions to support intuitive understanding could be more evident in females than in males. Moreover, subcortical network centrality predicted intuitive understanding in females but not in males. Consistently, a case report of a subcortical lesion with concurring emotional empathy deficits concerned a female patient [67]. Sex differences in empathy as well as in the underlying neural mechanisms were frequently reported. Such differences could be explained partly by cultural and societal factors but might have phylogenetic and ontogenetic biological roots as well [46]. Considering the role of the dorsal striatum in action-reward relationships [110,111] it might be speculated that intuitive understanding has been a cultural or evolutionary rewarding drive for social behavior in females more than in males.

### 4.5. Limitations and Conclusions

Our study has some limitations. Firstly, due to the importance of gender in the neural representation of empathic traits, it may be necessary to investigate other gender identities (i.e., non-cisgender) and perhaps also consider sexual orientations to draw a complete picture of the brain–behavior scaffold of empathy. Secondly, we investigated the intrinsic neural representation of empathy traits in young Italian adults, but it would be interesting to assess cultural and age effects related to our findings. Finally, although we carefully interpreted the behavioral relevance graph metrics, the empirical significance of many centrality measures is not always univocal and sometimes overlaps.

In conclusion, we showed that perceptual and cognitive dimensions of emotional empathy at a pre-reflective level could be associated with distinct functional brain networks. Whereas the vicarious experience trait was found to be associated with the fronto-parietal network, including higher-order sensorimotor circuits, the intuitive understanding trait could be traced to centrality measures of subcortical and lower-level sensorimotor networks. Moreover, distinct topological properties yielded different results for intuitive understanding, suggesting that specific brain modules may provide specific contributions to this empathic trait. Salience network regions were related to both vicarious experience and intuitive understanding. In line with the complexity of empathic processes and their dependence on multiple brain systems, empathic traits were predicted by inter-network connectivity indices rather than intra-network indices.

These results contribute to explaining the local neural signal propagation properties within the complex brain network underlying individual inclinations towards pre-reflective, intersubjective understanding. The findings also could help to understand the impact of lesions and stimulation of the identified regions and networks on trait empathy as depending on the integrity of a complex system in addition to focal brain processes.

## Figures and Tables

**Figure 1 brainsci-12-00477-f001:**
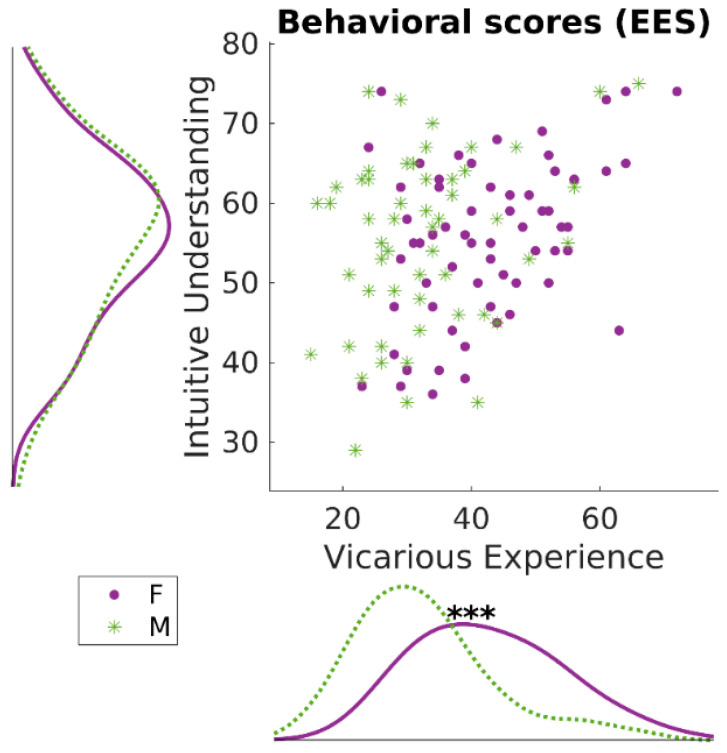
Behavioral Results. The two subscales of the Empathic Experience Scale were moderately positively associated (*r* = 0.315). Furthermore, while intuitive understanding scores were equal across females and males (*p* = 0.99), vicarious experience scores were higher in females (*p* < 0.001: ***), confirming the results from previous studies [13].

**Figure 2 brainsci-12-00477-f002:**
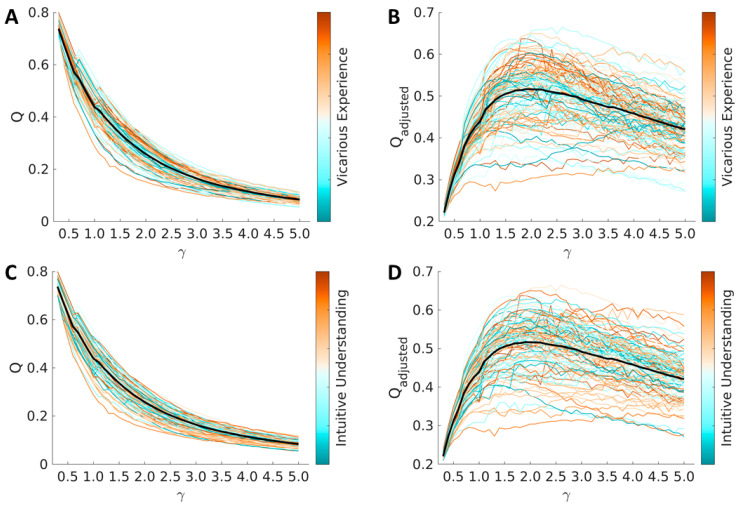
Modularity (**A**,**C**) and adjusted modularity (**B**,**D**) are represented both with average values (thick black line) and with single-subject values (thin colored lines). In (**A**,**B**), increased values of vicarious understanding are represented with more intense green colors. In (**C**,**D**), increased values of intuitive understanding are represented with more intense purple colors. As the figures show, and as described in the main text, modularity was not associated with vicarious experience or intuitive understanding.

**Figure 3 brainsci-12-00477-f003:**
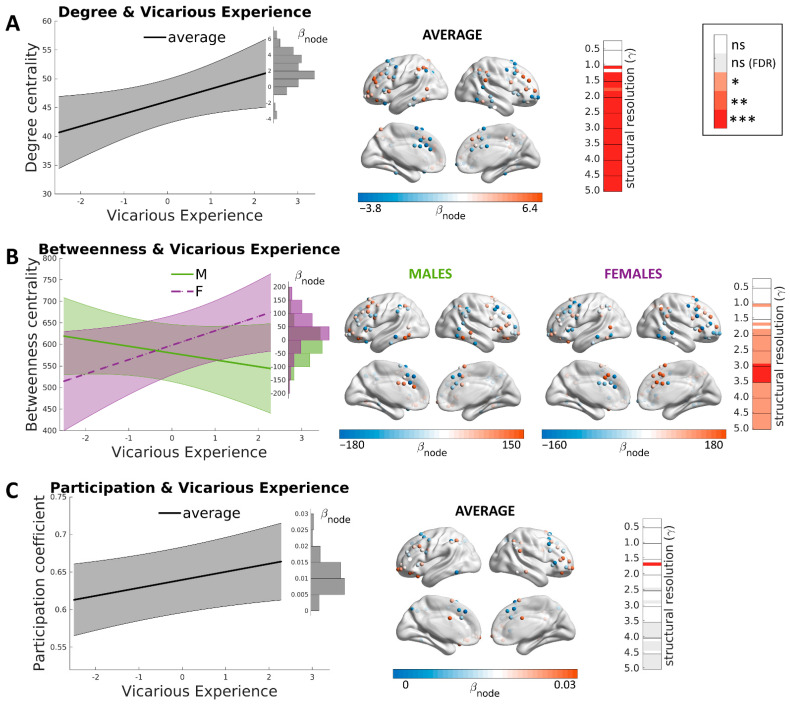
Brain correlates of Vicarious Experience. (**A**) The association between degree centrality of the fronto-parietal module and vicarious experience did not depend on sex, as shown in the prediction plot on the left. The highest contributors to the association were in the dorsolateral prefrontal cortex. The result was consistent with every value of structural resolution in which the fronto-parietal module was detected (γ > 1.0). (**B**) Betweenness centrality in the fronto-parietal module and vicarious understanding were positively associated in females and slightly negatively associated in males. The highest associations were in mid-cingulate regions. The result was consistent with medium and high structural resolutions (γ > 1.5). (**C**) The association with the participation coefficient of fronto-parietal nodes was not dependent on sex. Such a result was significant after multiple comparisons only with a specific value of structural resolution (γ > 1.7). Legend for significance with respect to structural resolutions: white = non-significant; gray = non-significant after correction for multiple comparisons; light red * = *p* < 0.05; medium red ** = *p* < 0.01; dark red *** = *p* < 0.005.

**Figure 4 brainsci-12-00477-f004:**
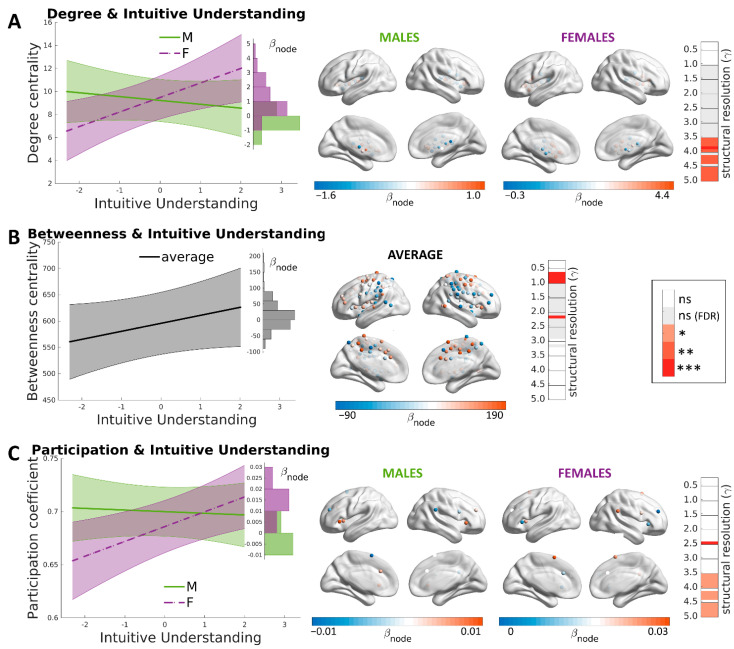
Brain correlates of Intuitive Understanding. (**A**) The association between degree centrality in the subcortical module and intuitive understanding was positive in females (magenta) and absent in males (green), as shown by prediction plots on the left. The middle section of the subfigure shows that the highest contributions to this association were particularly due to caudate and putamen regions. Such association was significant with high values of structural resolution, as shown on the right (γ > 3.5), since with lower values, subcortical nodes are merged in the same module with cerebellar nodes. (**B**) Betweenness centrality and intuitive understanding were positively associated in a sensorimotor module. The highest associations were in mid-cingulate regions. The result was consistent with low structural resolutions (0.5 < γ < 1.0). (**C**) The association with the participation coefficient of salience nodes was true only in females, and it was true with high values of structural resolutions (γ > 3.5), which allow smaller modules. Legend for significance with respect to structural resolutions: white = non-significant; gray = non-significant after correction for multiple comparisons; light red * = *p* < 0.05; medium red ** = *p* < 0.01; dark red *** = *p* < 0.005.

## Data Availability

Data available on request due to restrictions (e.g., privacy);The data presented in this study are available on request from the corresponding author.

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
