# Peer review of "Intrinsic Shapes of Empathy: Functional Brain Network Topology Encodes Intersubjective Experience and Awareness Traits"

_brainsci, 2022, doi:10.3390/brainsci12040477_

Round 1
Reviewer 1 Report
General Comments
This study aimed to investigate the relationship between intrinsic brain network architecture and psychometric measures (ESS) of pre-reflective intersubjective empathic traits. The paper is generally well written and structured. The results of this study are valuable for more understanding of the individual difference in empathy-related network integrity. However, there still had some minor parts could be improved.
The specific comments as below.
Introduction:
It should have more evidence to support your motivation of this study that why the perceptual and cognitive dimensions on a pre-reflective level have been relatively ignored.
It need to give a more detailed explanation that why you hypothesize the vicarious experience and intuitive understanding to be associated with connectivity profiles of distinct brain regions within empathy-related networks.
Method:
The procedure of how participants filled the Empathic Experience Scale should be described in detail when performed two resting-state runs for assessing the functional architectures of the brain.
Author Response
--------------------------------
REVIEWER 1
--------------------------------
This study aimed to investigate the relationship between intrinsic brain network architecture and psychometric measures (ESS) of pre-reflective intersubjective empathic traits. The paper is generally well written and structured. The results of this study are valuable for more understanding of the individual difference in empathy-related network integrity. However, there still had some minor parts could be improved.
The specific comments as below.
REVIEWER COMMENT
Introduction:
It should have more evidence to support your motivation of this study that why the perceptual and cognitive dimensions on a pre-reflective level have been relatively ignored.
It need to give a more detailed explanation that why you hypothesize the vicarious experience and intuitive understanding to be associated with connectivity profiles of distinct brain regions within empathy-related networks.
AUTHORS: We thank the Reviewer for this comment to improve the introduction section. The text of the revised version of the manuscript has been updated and below we indicate the main paragraphs that were modified to address the Reviewer’s concerns.
Introduction
Thus, whereas this literature, as well as previous studies on the brain correlates of individual differences in empathy traits (Christov-Moore et al. 2020; Esmenio et al. 2019; Winters et al. 2021; Cox et al. 2012), mainly addressed a distinction between more general emotional responses and cognitive/reflective aspects of empathy, the differentiation between perceptual and cognitive dimensions of empathy on a pre-reflective level (i.e., awareness without explicitly inferring others’ experiences) and their neural substrates have been relatively ignored (Innamorati et al. 2019).
To address these methodological, psychometric and theoretical issues,
…
Although emotional and cognitive empathy are considered distinct processes largely associated with differential brain circuits, interpersonal sharing and pre-reflective understanding are often considered intertwined processes. It is still unclear whether and how different brain systems have distinguishable roles in vicarious experience and intuitive understanding as parts of pre-reflective forms of empathy (Gallese 2005, 2014; Molnar-Szakacs and Uddin 2013). In accordance with evidence that vicarious experience and intuitive understanding reflect psychometrically independent dimensions (Innamorati et al., 2019), we hypothesize that individual differences in these distinct dimensions are associated with connectivity profiles of distinct brain regions within empathy-related networks.
…
Based on the above literature, the fronto-parietal network including the MNS implied in the empathic sharing of actions, emotions and sensations (Molenberghs et al., 2012; Keysers and Gazzola 2009) could be expected to explain individual differences in the vicarious experience trait. By contrast, the intuitive understanding trait partially resembles the functions ascribed to default mode, prefrontal and tempero-parietal junction regions (Shamay-Tsoory 2011; Zaki and Ochsner 2012), although it particularly differs from cognitive empathy through its emphasis on the absence of effortful inferential processes. Finally, the salience network could show overlapping associations with both vicarious experience and intuitive understanding constituting a common system for multiple empathic processes (Fan et al., 2011; Bernhardt and Singer, 2012).
…
Moreover, subcortical structures have been identified as main hub regions (Bell and Shine 2016; Oldham and Fornito 2019) with evolutionary importance in empathy (Decety et al. 2012; Northoff and Panksepp 2008), and their lesions likely interrupt networks relevant for the empathic recognition of emotions (Couto et al. 2013; Weddell 1994; Masterman and Cummings 1997). Hence, connectivity profiles of medial/lateral prefrontal cortices and subcortical structures could be particularly relevant for intuitive understanding traits.
REVIEWER COMMENT
Method:
The procedure of how participants filled the Empathic Experience Scale should be described in detail when performed two resting-state runs for assessing the functional architectures of the brain.
AUTHORS: We apologize for not reporting this information on the previous version of the manuscript, which is now reported in the updated version of the manuscript.
Materials and Methods
Empathic Experience Scale
All participants completed the EES in a quiet room on a different day than the fMRI scanning after receiving the following standardized instructions: “Please read very carefully the following statements and rate how strongly they describe how you normally feel. Use the following scale. Do not linger too much on the single statements and answer as sincerely as you can”.
Reviewer 2 Report
This project aims to explore the relationship between brain connectivity networks and such empathy traits as vicarious experience and intuitive understanding. The study found different brain network connectivity correlated with psychometric measures of emotional empathy and intuitive awareness of the emotions of others as well as individual differences in the brain mapping a function of gender. The study is timely and addresses an understudied but important topic.
p.3 4th paragraph from the top: Please spell out the direction of differences for the empathy traits among sexes: i.e. female versus male. More specifically, what a higher vicarious experience score in females would mean?
p.4 the first full paragraph from the top: Please spell out how do you conceive of vicarious experience versus intuitive awareness on the emotion-cognition dimension – from the rest of the paper it does not seem that vicarious experience =cognitive & intuitive awareness=emotions but at this point it might seem that that is the direction.
p.4 Participants: please report Hand dominance if it was taken to make sure that this is not the confound of the right hemisphere male and female differences.
Author Response
--------------------------------
REVIEWER 2
--------------------------------
This project aims to explore the relationship between brain connectivity networks and such empathy traits as vicarious experience and intuitive understanding. The study found different brain network connectivity correlated with psychometric measures of emotional empathy and intuitive awareness of the emotions of others as well as individual differences in the brain mapping a function of gender. The study is timely and addresses an understudied but important topic.
REVIEWER COMMENT
p.3 4th paragraph from the top: Please spell out the direction of differences for the empathy traits among sexes: i.e. female versus male. More specifically, what a higher vicarious experience score in females would mean?
AUTHORS: Thank you for this suggestion. We explain the meaning of a higher vicarious experience score in the following paragraphs.
Results
Behavioral results: Empathic Experience Scale
Instead, a significant difference was found when contrasting males vs. females with respect to their vicarious experience scores, with females showing higher scores (t114 = 4.85; p < .001; Confidence Intervals for mean difference [5.89, 14.03]). This suggests that females could have a stronger tendency than males to experience vicarious sensorimotor or emotional states when witnessing the emotional state of someone else, i.e. to participate stronger in other individuals’ emotional state by experiencing similar emotions.
Discussion
It could be hypothesized the tendency for a greater vicarious experience score among women could be explained in the light of their role as primary care-givers for children. This inclination may have favoured selective pressures for greater vicarious empathy in women with the benefit of reading the infant’s mental states and needs rapidly, thus promoting the infant’s survival (Bembich et al., 2022; Trevarthen 1979).
REVIEWER COMMENT
p.4 the first full paragraph from the top: Please spell out how do you conceive of vicarious experience versus intuitive awareness on the emotion-cognition dimension – from the rest of the paper it does not seem that vicarious experience =cognitive & intuitive awareness=emotions but at this point it might seem that that is the direction.
AUTHORS: Thank you for clarifying this issue. We modified the corresponding paragraph in the following way:
Introduction
Based on the above literature, the fronto-parietal network including the MNS implied in the empathic sharing of actions, emotions and sensations (Molenberghs et al., 2012; Keysers and Gazzola 2009) could be expected to explain individual differences in the vicarious experience trait. By contrast, the intuitive understanding trait partially resembles the functions ascribed to default mode, prefrontal and tempero-parietal junction regions (Shamay-Tsoory 2011; Zaki and Ochsner 2012), although it particularly differs from cognitive empathy through its emphasis on the absence of effortful inferential processes.
p.4 Participants: please report Hand dominance if it was taken to make sure that this is not the confound of the right hemisphere male and female differences.
AUTHORS: We thank the reviewer for this observation. Information about dominance is now reported in the manuscript. Given the scarcity of left-handed participant, and the balance across genders, we can claim that this effect is not a confound for the study (62 females (60 right-handed, 2 left-handed) and 54 males (52 right-handed, 2 left-handed).
Materials and Methods
Participants
116 healthy Italian adults (62 females of which 60 right-handed and 54 males of which 52 right-handed, aged 23 ± 3)…
Reviewer 3 Report
The authors have proposed a network approach to explore the psychological trait of empathy, the method is original and well argued. The statistical approach appears robust and the structure paper's has a good balance across the different scientific fields taken into consideration. However, the following technical issues should be considered:
- Section ‘MRI Data Preprocessing’, pg.5: methods about preprocessing steps are a bit concise and should be clarify. Anatomical and functional images should be differentiated i.e., slice timing is performed on functional images only while normalization on both the anatomical and the functional ones. Furthermore, since these preliminary steps are important for the result reproducibility, the clear temporal sequence of procedures should be described, as I understand the following procedures were used: time slicing, realignment, normalization and smoothing. Furthermore, why did you use a concatenation of different acquisition? Can you clarify how did you make the statistical model about it?
- Section ‘fMRI Data Analysis Network analysis’, pg.6: The proportional threshold used to build networks is 10%. Since the choice of such a value could change the results, it would be appropriate to justify this choice or showing other indices to describe matrices (such as minimum threshold to have a non-connected network or the corresponding minimum correlation value obtained after the thresholding).
- Section ‘Results’, Fig.2 and 3: The structural resolution (γ) is a useful parameter to explore the different architecture of modules. However, the corresponding Q values is not shown. Thus, to ensure a good partition of the network, such a value should be indicated for the each γ chosen (see Fortunato & Barthélemy, 2007). In addition, since different combination of nodes can be ascribed to different modules as a function of the γ parameter, a table containing the name of brain regions in the different modules should be inserted for each parameters value used. This could improve the anatomical understanding of the functional implications of the results.
- Section ‘Discussion’: about the network analysis, the authors used different centrality measures appropriately. In the section ‘Analysis of Neural Networks-Empathic Traits Associations’ (pg.6), a clear and concise description about each centrality measure is already inserted. However, their interpretation is not completely exhaustive in the discussion section. Though the participation coefficient was taken into consideration for the results interpretation, discussion on the other centrality measures should be expanded. Since, these methods characterize different dynamical properties, a clear interpretation about different results between centrality measures should be included. For a critical description of the network centrality see Borgatti 2004; 2005.
Bibliography
Fortunato, S, Barthélemy, M. Resolution limit in community detection. Proc Natl Acad Sci U S A. 2007;104(1):36-41.
Borgatti, Stephen, P. Centrality and Network Flow. Social Networks. 2005; 27: 55-71.
Borgatti, Stephen, P., Everett, Martin, G. A Graph-Theoretic Perspective on Centrality. Social Networks. 2006; 28(4): 466-484.
Author Response
--------------------------------
REVIEWER 3
--------------------------------
The authors have proposed a network approach to explore the psychological trait of empathy, the method is original and well argued. The statistical approach appears robust and the structure paper's has a good balance across the different scientific fields taken into consideration. However, the following technical issues should be considered:
REVIEWER COMMENT
- Section ‘MRI Data Preprocessing’, pg.5: methods about preprocessing steps are a bit concise and should be clarify. Anatomical and functional images should be differentiated i.e., slice timing is performed on functional images only while normalization on both the anatomical and the functional ones. Furthermore, since these preliminary steps are important for the result reproducibility, the clear temporal sequence of procedures should be described, as I understand the following procedures were used: time slicing, realignment, normalization and smoothing. Furthermore, why did you use a concatenation of different acquisition? Can you clarify how did you make the statistical model about it?
AUTHORS: We thank the reviewer for this comment and we apologize for having reported our preprocessing of MRI and fMRI images incompletely or not clearly. In the revised version of the manuscript, we improved our description of the procedure used to preprocess MRI data and we made clear the temporal sequence of the preprocessing procedures implemented.
To answer to the second part of the reviewer’s observation, the two acquisitions were consecutive. It is a very common procedure to include many data points for the estimation of functional connectivity using more short runs, thus without stressing the experimental subjects with very long runs. In fact, a 15-minutes long resting-state acquisition would be more affected by movements and the subject is more likely to fall asleep in the scanner. Instead, using two shorter runs allows to keep the same amount of data reducing artifact and eventual “state” bias (i.e., sleeping).
Materials and Methods
MRI Data Preprocessing
For each subject, the two resting state runs were concatenated. The resulting functional images were despiked, deobliqued and time-shifted to obtain slices with the same acquisition timing. The high-resolution anatomic image (T1) was skull-stripped, aligned to the Montreal Neurological Institute (MNI) template, motion-corrected using a six-parameter rigid body realignment, and then warped to the standard space through a non-linear high-efficiency procedure. Thus, functional images were aligned to the standard space and to the T1 image using automatically optimized non-linear warping algorithms. During preprocessing, motion parameters for each participant were stored to be used as regressor of non-interest in further steps. The resulting functional images were blurred using a Gaussian smoothing kernel with 5 millimeters at its full-windows half maximum. Finally, fMRI timeseries were scaled to obtain an average value of 100, a procedure essential to translate the BOLD signal to a "percent signal change", which is a more meaningful measure (Chen et al., 2017). The quality of the preprocessing was assessed both using AFNI built-in quality check programs and utilities and through the direct examination of the preprocessed images.
REVIEWER COMMENT
- Section ‘fMRI Data Analysis Network analysis’, pg.6: The proportional threshold used to build networks is 10%. Since the choice of such a value could change the results, it would be appropriate to justify this choice or showing other indices to describe matrices (such as minimum threshold to have a non-connected network or the corresponding minimum correlation value obtained after the thresholding).
AUTHORS: We adopted a common proportional thresholding procedure which has been shown to remove spurious connectional values (Achard and Bullmore, 2007; van den Heuvel et al., 2017) and allows to obtain sparsely connected matrices at the single-subject level with the aim of keeping the total number of connections fixed across all individuals. Such approach controls for the influence of network density on the computation and comparison of graph metrics across individuals (van den Heuevl et al., 2008; Ginestet et al., 2011). This information is now reported in the manuscript.
fMRI Data Analysis
Binary undirected graphs were built from connectomes (connectivity matrices) after a thresholding procedure in which the 10% among the strongest connections were maintained to remove spurious connectional values (Achard and Bullmore, 2007; van den Heuvel et al., 2017) and to obtain sparsely connected matrices at the single-subject level with the aim of keeping the total number of connections fixed across all individuals. Such approach controls for the influence of network density on the computation and comparison of graph metrics across individuals (van den Heuevl et al., 2008; Ginestet et al., 2011).
REVIEWER COMMENT
- Section ‘Results’, Fig.2 and 3: The structural resolution (γ) is a useful parameter to explore the different architecture of modules. However, the corresponding Q values is not shown. Thus, to ensure a good partition of the network, such a value should be indicated for the each γ chosen (see Fortunato & Barthélemy, 2007). In addition, since different combination of nodes can be ascribed to different modules as a function of the γ parameter, a table containing the name of brain regions in the different modules should be inserted for each parameters value used. This could improve the anatomical understanding of the functional implications of the results.
AUTHORS: We thank the reviewer for this observation and we agree that it is important to show modularity values. We report this information now in the new Figure 2. Additionally, we tested if these values were correlated with the empathic traits of interest. However, global modularity was not associated with vicarious experience or intuitive understanding, indicating that the relationship between empathy and functional connectivity is not “global” but more dependent on specific modules. With respect to the testing of a good partition, we would like to note that the brain probably works simultaneously at multiple scales, so that it is essential to investigate brain-behavioral coding at the cross-γ level. However, we only tested modules which were significant after the Newman-Girvan procedure to exclude inconsistent/small modules.
Materials and Methods
fMRI Data Analysis
Values of modularity (Q) were extracted for each individual at each level of structural resolution investigated and reported (Fortunato & Barthélemy, 2007). Using linear Pearson correlation, we tested if modularity was associated with vicarious experience and intuitive understanding, possibly reflecting a “global” encoding of empathic traits in the brain connectome.
Results
Global Modularity
Average and single-subject values of modularity (Q) across values of structural resolution (γ) are reported in Figure 2. As expected, values of optimized modularity decreased with increasing values of structural resolution (smaller modules). These values are consistent with the literature (see Di Plinio et al., 2020). Modularity was not associated with vicarious experience (average cross-γ Person correlation = 0.014+-0.049, p values < 0.30) nor it was with intuitive understanding (average cross- γ Person correlation = -0.066+-0.022, p values < 0.30).
Figure 2. Modularity (A, C) and adjusted modularity (B, D) are represented both with average values (thick black line) and with single-subject values (thin colored lines). In A and B, increased values of vicarious understanding are represented with more intense green colors. In C and D, increased values of intuitive understanding are represented with more intense purple colors. As the figures show, and as described in the main text, modularity was not associated with vicarious experience or intuitive understanding.
Results
Brain Correlates of Vicarious Experience
Regions of interest and their coordinates in the MNI space with respect to the brain correlates of vicarious experiences are reported in the supplementary file (tables S1-S3).
…
Brain Correlates of Intuitive Understanding
Regions of interest and their coordinates in the MNI space with respect to the brain correlates of vicarious experiences are reported in the supplementary file (tables S4-S6).
REVIEWER COMMENT
- Section ‘Discussion’: about the network analysis, the authors used different centrality measures appropriately. In the section ‘Analysis of Neural Networks-Empathic Traits Associations’ (pg.6), a clear and concise description about each centrality measure is already inserted. However, their interpretation is not completely exhaustive in the discussion section. Though the participation coefficient was taken into consideration for the results interpretation, discussion on the other centrality measures should be expanded. Since, these methods characterize different dynamical properties, a clear interpretation about different results between centrality measures should be included. For a critical description of the network centrality see Borgatti 2004; 2005.
AUTHORS: We agree with the reviewer that the interpretation of the findings related to centrality should have been exposed more clearly in the Discussion, although it is not easy to concisely describe the whole pattern of findings without being too wordy. In the revised manuscript, we added a more detailed description of our findings, especially with respect to degree and betweenness centrality. We also added the useful references suggested by the reviewer.
Introduction
Centrality measures indicate the importance of a node or a set of nodes for communication performance within a network (Borgatti, 2005; Borgatti and Everett, 2006). Such indices are relevant to obtain insights, for example, on the intrinsic neural basis of empathy as well as on the impact of lesions on empathy, privileging the network perspective, rather than localized functions. Since centrality can be defined in various ways, three commonly used centrality measures were used here to quantify the topological importance of brain regions. Firstly, degree centrality is determined by the number of connections of each node. Secondly, betweenness centrality is based on the number of shortest paths between pairs of nodes that pass through a determined node. Thirdly, closeness centrality depends on the average length of shortest paths between a node and all the other network nodes. Using many complementary measures of centrality allows the thorough description of nodal (and modular) contribution to the overall information flow within a network (Friedkin, 1991).
Discussion
Brain Networks and Empathy Concepts
…
One the one hand, the association of vicarious experience with two centrality measures in the fronto-parietal system indicate a multifaceted involvement of these fronto-parietal nodes in processes of self-other resonance, encompassing both radial (degree) and medial (betweenness) information flow (Borgatti and Everett, 2006; Friedman, 1991). In fact, these nodes have a role both in the propagation of information across subsystems (probably through their high inter-network connectivity, see Di Plinio and Ebisch , 2018) and in the high-order modulation of socioemotional and affective facets of precepted stimuli (Gratton et al. 2012; Hillis, 2014; Aerts et al. 2016). On the other hand, intuitive understanding relied on different brain systems. Thus, the natural awareness of others’ feelings is influenced simultaneously by the connectional strength (degree) of subcortical regions and by the connectional design of sensorimotor regions, supporting the importance of both systems in empathic processes (Couto et al. 2013; Weddell 1994; Keysers et al. 2010; Christov-Moore et al. 2020).
Importantly, the involvement of salience network regions’ betweenness in both empathic dimensions (vicarious experience and intuitive understanding) indicate a convergent engagement of these brain regions in self-other matching processes including experiencing as well as understanding of one’s own and others’ emotions and feelings (Xu et al. 2020; Yao et al. 2016; Bird et al. 2010; Hogeveen et al. 2016; Lamm et al. 2011). The anterior insula has been associated with different levels of self-processing from interoceptive to exteroceptive and mental levels (Qin et al. 2020), showing higher resting state centrality indices for all three levels of self (Scalabrini et al. 2021). Our findings imply that the anterior insula could be a key hub for the propagation of general viscero-motor information necessary for sharing as well as understanding others’ emotional states (Gallese et al. 2004; Fan et al. 2011). Removing anterior insular or dorsal cingulate nodes from the network would impair the merging of self-related information with the representation of other’s feelings which is necessary to bodily resonance and pre-reflective understanding of others. These findings agree and help to explain behavioral disturbances highlighted in lesion studies of anterior insula and cingulate cortex, like alexithymia (Chau et al., 2018), aberrant interpersonal behavior (Belfi et al., 2015), and decreased social interactions (Hadland et al., 2003). Overall, we further extend the functional role of the insula, which could act as a key hub not only for self-processing (subjectivity), but also for empathic processing (intersubjectivity).